# Antioxidant and Anti-Inflammatory Activities of *Sargassum macrocarpum* Extracts

**DOI:** 10.3390/antiox11122483

**Published:** 2022-12-16

**Authors:** Hoon Kim, Hyun Young Shin, Eun-Jin Jeong, Hak-Dong Lee, Ki Cheol Hwang, Kwang-Won Yu, Sullim Lee, Sanghyun Lee

**Affiliations:** 1Department of Food Science and Nutrition, Chung-Ang University, Anseong 17546, Republic of Korea; 2Department of Integrated Biomedical and Life Science, Korea University, Seoul 02841, Republic of Korea; 3Department of Food and Nutrition, Korea National University of Transportation, Jeungpyeong 27909, Republic of Korea; 4Department of Plant Science and Technology, Chung-Ang University, Anseong 17546, Republic of Korea; 5Venture Research Center, Rafarophe Co., Cheongju 28164, Republic of Korea; 6Department of Life Science, Gachon University, Seongnam 13120, Republic of Korea; 7Natural Product Institute of Science and Technology, Anseong 17546, Republic of Korea

**Keywords:** HaCaT, RAW 264.7, brown seaweed, anti-dermatitis activity, fucose-containing polysaccharide

## Abstract

Oxidative stress and the inflammatory response are known to be the most important pathological factors for aging skin cells. Therefore, substances that protect skin cells from oxidative stress and inflammatory reactions of the skin have potential as functional ingredients for skin care. In the present study, we investigated the potential of *Sargassum macrocarpum* as an anti-inflammatory candidate for inflammatory skin disease. Antioxidant and anti-inflammatory activities are desirable properties in such functional materials. The total polyphenol content as well as antioxidant and anti-inflammatory activities were evaluated in hot-water (HES) and ethanol (EES) extracts of *S. macrocarpum*. The polyphenol content was higher in the HES (HES: 115.9 ± 15.3 mg GA/g, EES: 3.9 ± 0.5 mg GA/g), and the HES also had ABTS (HES: IC_50_ 1.0 ± 0.0 mg/mL, EES: IC_50_ 16.09 ± 0.7 mg/mL) and DPPH (HES: IC_50_ 6.50 ± 0.3 mg/mL, EES: IC_50_ 35.3 ± 3.1 mg/mL) radical scavenging capacities as well as FRAP activity (HES: IC_50_ 18.8 ± 0.4 mg/mL, EES: IC_50_ n.d.). Compared with the EES at the equivalent concentration range (1.25–20 μg/mL), the HES exerted a more potent inhibitory activity on LPS-stimulated nitric oxide (10.3–43.1%), IL-6 (15.7–45.0%), and TNF-α (14.1–20.8%) in RAW 264.7 macrophage cells in addition to TNF-α and IFN-γ-facilitated IL-6 (10.9–84.1%) and IL-8 (7.7–73.2%) in HaCaT keratinocytes. These results suggested that water-soluble materials might be deeply involved in the antioxidant and anti-inflammatory activity in *S. macrocarpum*. General composition analysis indicated that the HES contains more carbohydrates and polyphenols than the EES, and the monosaccharide composition analysis suggested that fucose-containing sulfated polysaccharide and β-glucan might be potent anti-inflammatory candidates in the HES. The present study presents important preliminary results and a valuable strategy for developing novel anti-skin dermatitis candidates using a hot-water extract of *S. macrocarpum*.

## 1. Introduction

Skin cosmetics contain multiple heavy metals that can damage the skin. The original goal of cosmetics was to conceal blemishes on the skin and appear beautiful. However, the skin should be protected from pathogens, such as bacteria and viruses, as well as external physical stimuli [1,2,3]. That is, cosmetics should not only create the appearance of beauty, but also maintain healthy skin. Typical examples include functional cosmetics and inner-beauty products. Functional cosmetics are applied directly onto the skin and aim to create a healthy effect on the skin, whereas inner beauty aims to improve the appearance of the skin through the ingestion of healthy foods.

One of the most important features of a substance that is ingested to improve the skin from within is its antioxidant capacity, that is, the ability to scavenge free radicals. All cells generate reactive oxygen species (ROS) during the process of energy production [4,5] and these free radicals are essential for cellular functions, such as the immune response, the regulation of cell differentiation, and the metabolism [6,7]. However, the free radicals produced by inflammatory reactions and an excessive UV exposure can lead to immune disorders, carcinogenesis, and cellular senescence [7].

Reactive oxygen species in the skin play major roles in the formation of wrinkles [8], which result from a reduced collagen synthesis in the dermal layer of the skin and accelerated collagen disintegration [9]. The matrix metalloproteinases (MMPs) degrade collagen, and the inflammatory inducers ROS, interleukin (IL)-1, and IL-6, as well as lipid peroxide which produces MMP that accelerates the formation of wrinkles [10]. Therefore, inflammatory indicators play an important role in determining inner beauty.

Many natural product extracts exert anti-inflammatory and antioxidant effects. Therefore, we investigated the antioxidant, cytotoxic, and anti-inflammatory properties of *Sargassum macrocarpum* extract. *S. macrocarpum* is a perennial brown algal seaweed distributed along the west coast of the North Pacific Ocean, and the southern coast of Korea and Jeju Island [11,12]. The physiologically active substances of brown algae include polysaccharides, phenolic compounds, phlorotannins, terpenoids, and steroids. The cell walls of brown algae mainly comprise fucoidan, alginate, laminarin, and their derivatives. Fucoidan is a representative active ingredient of brown algae that has anticancer and anti-inflammatory activities [13,14]. Nineteen *Sargassum* species can inhibit nitric oxide and lipid accumulation, and they have an antioxidant capacity [15]. *S. serratifolium* extract reduces neuroinflammation in mice [16]. *S. fulvellum* and *S. thunbergii* exert antipyretic, analgesic, and anti-inflammatory activities in mice [17] and sargachromanol G isolated from *S. siliquastrum* exerts anti-inflammatory effects in RAW 264.7 cells [18]. Although various biological activities have been identified in various *Sargassum* species, little is known about the properties of *S. macrocarpum*. According to a previous study, the two Italian varieties of common bean showed a potent antioxidant activity and anti-inflammatory activity depending on the concentration of the major phenols in them [19]. In addition, another study reported that the extracts of *A. macrostachyum*, *H. portulacoids*, and *S. europaea* showed different enzyme inhibition effects, with different total phenols and flavonoids depending on the extraction method [20]. As such, the composition, the content of the phytochemical, and bioactivity may be different depending on the types of the variety used and the extraction method. Therefore, we investigated the feasibility of using *S. macrocarpum* as an inner-beauty material.

## 2. Materials and Methods

### 2.1. Plant Materials

Fresh *S. macrocarpum* collected from the coastal area of Jeju Island (the Republic of Korea) during 2021 (Figure 1) was washed with distilled water and dried under sunlight. A voucher specimen was deposited at the Rafarophe Co., the Republic of Korea.

### 2.2. Preparation of Ethanol and Water Extracts of S. macrocarpum

Dried *S. macrocarpum* (100 g) was immersed in 500 mL of 95% ethanol (EtOH) (Samchun Chemicals, Seoul, the Republic of Korea) and stored at 25 °C for three days. Dried *S. macrocarpum* (100 g) was also heated in 4 L of purified water at 107 °C for 4 h in an industrial extractor (Daehan Median Co., Gunpo, the Republic of Korea). Both extracts were passed through a polyester cloth filter (20 μm; Hyundai Micro, Anseong, Korea), rotary evaporated (Buchi Korea Inc., Gwangmyeong, the Republic of Korea), and lyophilized (Ilshin Biobase, Daejeon, the Republic of Korea) to obtain the EtOH (EES) and hot-water (HES) *S. macrocarpum* extracts. The extraction yields of the EES and HES were 8.8% and 21.2% against the dried material, respectively.

### 2.3. Assessment of Antioxidant Activities Using ABTS, DPPH, and FRAP

We compared the antioxidant activities of the extracts using 2,2′-azino-bis(3-ethylbenzothiazoline-6-sulfonic acid) (ABTS), 2,2-diphenyl-1-picrylhydrazyl (DPPH) (both from Sigma-Aldrich, St. Louis, MO, USA), and ferric reducing anti-oxidant power (FRAP) as described [21], with l-Ascorbic acid (Sigma-Aldrich) as the reference. The antioxidant activity is expressed as the ascorbic acid equivalent antioxidant capacity (AEAC) in the mg/g extract.

### 2.4. Assessment of Total Polyphenol and Flavonoid Content

The total polyphenol contents were measured by reacting extracts (10 μL) with 10 μL of 2% Na_2_CO_3_ and 200 μL of 50% Folin–Ciocalteu reagent for 30 min. The absorbance at 750 nm was measured using a microplate reader (Tecan Group AG., Männedorf, Switzerland). The total polyphenol contents are shown as gallic acid equivalents (GAE) mg/g extract determined using a gallic acid calibration curve. The total flavonoid content was determined as described [22] with a slight modification. The extract (20 μL) in 80% ethanol (40 μL) was reacted with 10% aluminum nitrate (20 μL), 1 M potassium acetate (20 μL), and 80% ethanol for 10 min, then the absorbance at 415 nm was measured. The total flavonoid contents were calculated as the quercetin equivalents (QE) mg/g sample using a quercetin calibration curve.

### 2.5. Analysis of General Composition and Sugars

The total sugar, uronic acid, and polyphenol contents were determined using phenol-sulfuric acid [23], *m*-hydroxibiphenyl [24], and Folin–Ciocalteu [25] methods and the standard references, galactose (Gal), galacturonic acid (GalA), and gallic acid (GA).

### 2.6. Cytotoxicity and Anti-Inflammatory Activity in RAW 264.7 Cells

RAW 264.7 cells (Korean Cell Line Bank [KCLB], Seoul, the Republic of Korea) were incubated at 37 °C under a humidified 5% CO_2_ atmosphere in Dulbecco’s modified Eagle’s medium (DMEM), supplemented with 10% fetal bovine serum (FBS) and 1% antibiotics (100 units/mL of penicillin and 100 µg/mL of streptomycin (all from Thermo Fisher Scientific Inc., Waltham, MA, USA). The cells (2 × 10^5^/mL) were seeded in 96-well plates (200 µL) and allowed to adhere. The medium was replaced with a serum-free medium containing various concentrations of extracts for 30 min, then the cells were induced with 10 µg/mL of *Escherichia coli* lipopolysaccharide (LPS; Sigma-Aldrich) for 24 h. The cells’ viability was assayed using 3-(4,5-dimethylthiazole-2-yl)-2,5-diphenyltetrazolium bromide (MTT; Life Technologies Co., Eugene, OR, USA). Formazan salts were solubilized in 100 μL of DMSO, then the absorbance at 550 nm was analyzed using an Epoch microplate reader (BioTek Instruments, Inc., Winooski, VT, USA). The nitric oxide (NO) in cell supernatants was measured using the Griess reagent (Thermo Fisher Scientific Inc.). The cell supernatants were homogenized and incubated with the Griess reagent at 25 ± 5℃ for 10 min, then the nitrite content was analyzed using an Epoch microplate reader (BioTek Instruments, Inc., Winooski, VT, USA) at a wavelength of 540 nm. The levels of tumor necrosis factor-alpha (TNF-α) and interleukin-6 (IL-6) in the cell supernatants were measured using mouse TNF-α (Thermo Fisher Scientific Inc.) and IL-6 (BD Biosciences, San Diego, CA, USA) ELISA kits, as described by the respective manufacturers.

### 2.7. Cytotoxicity and Anti-Inflammatory Activity Assessment in HaCaT Cells

HaCaT cells (CLS Cell Line Service GmbH, Eppelheim, Germany) were incubated in a DMEM supplemented with 10% FBS and 1% antibiotics at 37 °C under a humidified 5% CO_2_ atmosphere. The cells (1 × 10^5^/mL) were seeded in 96-well plates (200 µL) and allowed to adhere. Thereafter, the cells were incubated with a serum-free medium containing various concentrations of extracts for 1 h. Thereafter, 10 ng/mL each of the recombinant TNF-α and IFN-γ (both from R&D Systems, Minneapolis, MN, USA) was added to the cells for 24 h to induce inflammation. The cells’ viability was determined using MTT assays. The levels of IL-6 and IL-8 in the cell supernatants were measured using human ELISA kits (BD Biosciences) as described by the manufacturer.

### 2.8. Monosaccharide Composition Analysis

To analyze the monosaccharide composition, the sample was first hydrolyzed with 2 M of trifluoroacetic acid (Sigma-Aldrich) and then derived with 1-phenyl-3-methyl-5-pyrazolone (PMP; Sigma-Aldrich) to obtain the PMP-conjugated monosaccharide derivatives [24]. The PMP-monosaccharide derivatives were analyzed using the HPLC system (YL9100; Young Lin Co., Ltd., Anyang, the Republic of Korea) coupled with a UV detector (Young Lin Co., Ltd.) and YMC Triart C18 column (250 × 4.6 mm, 5 μm; YMC Co., Ltd., Kyoto, Japan). The mixed solvents comprising phosphate buffer (pH 6.7) and acetonitrile at a ratio of 83:17 was used as a mobile phase at a flow rate of 1 mL/min [26,27]. Eight standard references, pentoses (arabinose and xylose), hexoses (mannose, glucose, and galactose), methylated sugars (rhamnose and fucose), and acidic sugars (uronic acids; glucuronic acids; and galacturonic acids) were obtained from Sigma-Aldrich. The monosaccharide composition of the sample was presented as the molar percentage calculated from the peak area, molecular weight, and the response factor of each monosaccharide on the UVD.

### 2.9. Statistical Analysis

All the data were statistically analyzed by Student’s *t*-tests or a one-way ANOVA followed by Tukey’s post hoc tests using Predictive Analytics Software (PASW^®^) Statistics 18 (IBM Co., Armonk, NY, USA). The results are expressed as the means ± standard deviation (SD). Values with *p* < 0.05 were considered to be statistically significant.

## 3. Results and Discussion

We analyzed the antioxidant and anti-inflammatory activities of the EES and HES from *S. macrocarpum* to determine whether *S. macrocarpum* could serve as a functional agricultural product for the skin and to expand farm income. The total polyphenol, antioxidant, and anti-inflammatory activities of the HES and EES were evaluated.

### 3.1. Comparison of Antioxidant Activity

Table 1 shows the free radical scavenging activities determined using ABTS, DPPH, and FRAP assays. The radical scavenging of ABTS was more potent for the HES than the EES (IC_50_: 0.9 vs. 16.9 mg/mL) and DPPH (IC_50_: 6.5 vs. 30.1 mg/mL). Furthermore, the HES had 18.8 ± 0.4 AA mg/g FRAP activity. Therefore, the HES is a good radical scavenger and has a reducing power.

### 3.2. General Composition of S. macrocarpum Extracts

Phenolic compounds such as tannin, procyanidin, flavonoids, and phenolic acids are the most abundant secondary metabolites with a biological activity in plants [28,29,30]. Numerous phenolic hydroxyl groups attached to the aromatic ring structure easily bind to proteins, enzymes, and other macromolecules and hence exert antibacterial, antioxidant, and anti-inflammatory effects [29,31]. Bioactivities tend to increase in proportion to the phenolic content of plants. Therefore, the phenol content of plants can serve as basic data to evaluate the antioxidant activity of natural products [32,33]. Table 2 shows the contents of the total sugar, uronic acid, proteins, and the total polyphenols in the HES and EES. The HES was comprised mainly of total sugars (127.6 mg/g) and polyphenols (115.9 mg/g), with small amounts of proteins (41.8 mg/mL) and uronic acid (29.5 mg/g). In contrast, the EES contained 6.5, 2.5, 10.3, and 3.9 mg/g of the total sugar, uronic acid, proteins, and polyphenols, respectively. These results showed that the HES contains more constituents with antioxidant and anti-inflammatory effects than the EES.

### 3.3. Anti-Inflammatory Effects of S. macrocarpum Extracts in LPS-Induced RAW 264.7 Cells

Macrophages are the main cellular energy sources involved in inflammatory responses. They initiate the process of inflammation and trigger important defense mechanisms against tissue damage and foreign pathogens [34,35]. When tissue is damaged or infected, macrophages induce an inflammatory response by secreting infectious mediators, such as TNF-α, IL-6, and NO [35]. Therefore, inhibitors of pro-inflammatory mediators might be potential candidates for anti-inflammatory therapeutics. We investigated the effects of *S. macrocarpum* extracts on the production of pro-inflammatory NO and IL-6, and the possibility of their application as anti-inflammatory agents. We assessed the non-toxic concentration range of the HES and EES in RAW 264.7 cells. The viability of the cells incubated with 1.25–20 μg/mL of HES and EES was >85% compared with the LPS control (Figure 2A). Figure 3A shows the ability of the HES and EES to inhibit NO in RAW 264.7 cells with inflammation induced by LPS compared with dexamethasone (50 μg/mL) as the positive control (PC). Dexamethasone significantly decreased the amount of NO generated in the cells stimulated with LPS by 55.9% (32.6 vs. 14.4 μM), and 1.25–20 μg/mL of the concentration of the HES dependently decreased it by 10.3% to 43.1%). In contrast, the EES had no inhibitory ability.

### 3.4. Anti-Inflammatory Effects of S. macrocarpum Extracts in T + I-Induced HaCaT Cells

Based on these findings, we investigated the anti-dermatitis effects of *S. macrocarpum* extract on HaCaT cells, which are skin keratinocytes. Both TNF-α and IFN-γ (T + I) increase the expression of the pro-inflammatory cytokines IL-6 and IL-8 in keratinocytes and contribute to various skin inflammatory symptoms and diseases [36]. We confirmed that the non-toxic concentration range of the HES and EES was 1.25–20 μg/mL in HaCaT cells, the viability of which was >100% compared with the T + I group (Figure 2B). The cell proliferation activity was significantly higher in the 2.5–20 μg/mL HES than the T + I group (15.3%–33.6%).

Figure 4A,B compares the ability of the HES and EES extracts to inhibit IL-6 and IL-8 activity in HaCaT cells with that of PC cells incubated with 50 μg/mL dexamethasone. The assessment of the IL-6 inhibitory activity (Figure 4A) revealed that the PC group produced significantly less IL-6 than the T + I group (181.1 vs. 446.1 pg/mL; 59.4% inhibition). Figure 4A also indicates that the inhibitory effect of the T + I-induced IL-6 production was much higher in the group treated with HES (10.9–84.1% inhibition) compared to that with EES (2.6–17.5% inhibition) at the equivalent concentration ranges. The results showed that the HES inhibited IL-6 more effectively than the EES at the same concentration range. Figure 4B shows that less IL-8 was produced by the PC than the T + I group (169.2 vs. 583.8 pg/mL; 71.0% inhibition). The production of IL-8 was significantly suppressed by 1.25–20 μg/mL of HES to 7.7%–73.2% compared with the T + I group. At 1.25–10 μg/mL, the EES also significantly inhibited the production of IL-8, but 20 μg/mL showed an increased efficacy. These results show that the HES was more inhibitory against the activity of IL-8 than the EES.

### 3.5. Monosaccharide Composition of HES

From the above anti-inflammatory results, we concluded that the HES could exert a higher anti-inflammatory activity than the EES. We also speculated that the activity might be associated with the presence of carbohydrate units, total sugars, and uronic acid, as shown in Table 2. A previous study reported that fucose-containing sulfated polysaccharide (FCSP) and β-glucan, alginic acid, and polyphenolic compounds are the promising anti-inflammatory candidates of the water-soluble components in *Sargassum* sp. [37]. Thus, we aimed to further analyze the monosaccharide composition of the HES. The results were provided as an HPLC chromatogram (Figure 5) and molar percentages (Table 3) of the monosaccharides based on the peak areas on the chromatogram. 

As shown in Figure 5, the HES possessed high amounts of glucose and fucose units (35.7 and 34.3 mole%, respectively) in addition to small amounts of galactose (5.7 mole%), mannose (5.0 mole%), and glucuronic acid (4.9 mole%). In consideration of the fact that the term FCSP is now considered to be more proper than the old term fucoidan, because it bears various side chains, such as galactose, rhamnose, mannose, xylose, and glucose [38], the monosaccharide composition of the HES would be the structural properties of well-known algae polysaccharides, FCSP, and β-glucan. Nevertheless, it is necessary to conduct further chemico-structural investigations, using infrared spectroscopy and mass spectrometry, to clearly elucidate the structural property of the HES.

The possibility of using *S. macrocarpum* as a skin-functional ingredient was investigated. The contents of various components were higher, and the antioxidant effects were more potent in the HES than in the EES. An evaluation using macrophages and keratinocytes revealed more powerful anti-inflammatory effects in the HES than in the EES. These results are consistent with the fact that the HES is rich in constituents with antioxidant and anti-inflammatory potential. We confirmed that *S. macrocarpum* can serve as a functional raw material on an industrial scale. However, this study is limited because we assessed the potential antioxidant and anti-inflammatory activities of the HES only in HaCaT cells. Therefore, a further investigation of the various cell lines, such as normal skin fibroblasts and melanocytes, is needed before the HES can be developed as a functional skin material.

## Figures and Tables

**Figure 1 antioxidants-11-02483-f001:**
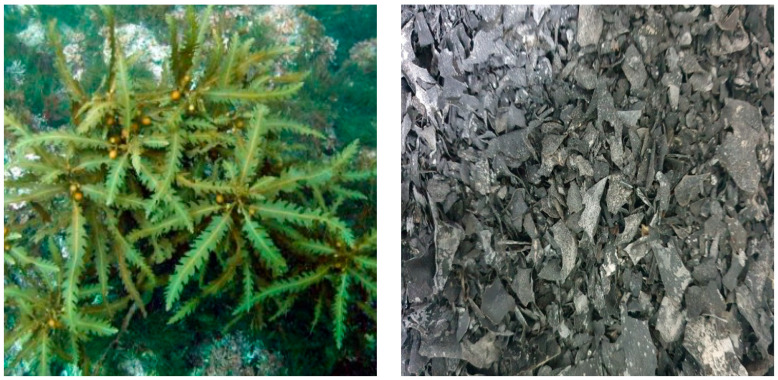
Representative image of wild (**left**) and dried (**right**) *S. macrocarpum*.

**Figure 2 antioxidants-11-02483-f002:**
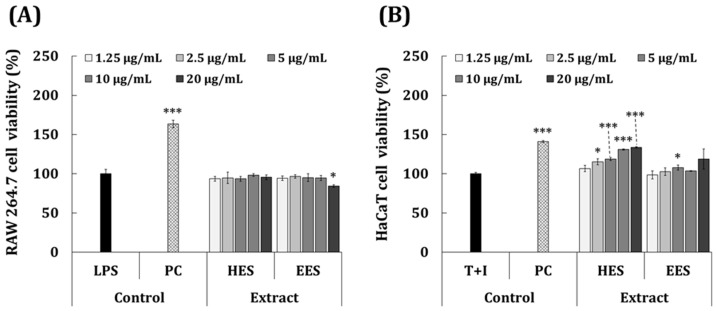
Effects of HES and EES on cell viability. (**A**) Lipopolysaccharide-induced RAW 264.7 cells and (**B**) T + I-induced HaCaT cells incubated with 2.5–20 μg/mL of EES or HES for 24 h. Cytotoxic effects were determined as cell viability using MTT assays. * *p* < 0.05; *** *p* < 0.001 vs. LPS or T + I (inflammation-induced) controls; Student’s *t*-test). EES, ethanol extract of *S. macrocarpum*; HES, hot-water extract of *S. macrocarpum*; PC, positive control (dexamethasone); T + I, tumor necrosis factor alpha + interferon-gamma.

**Figure 3 antioxidants-11-02483-f003:**
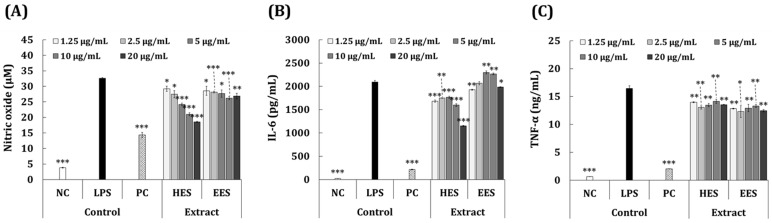
The ability of HES and EES to inhibit NO, IL-6, and TNF-α in LPS-induced RAW 264.7 cells. RAW 264.7 cells were incubated with 2.5–20 μg/mL of EES or HES for 24 h, then levels of (**A**) NO (**B**) IL-6, and (**C**) TNF-α were determined using Griess assays and ELISAs, respectively. * *p* < 0.05; ** *p* < 0.01; *** *p* < 0.001 vs. LPS (inflammation-induced) control (Student’s *t*-test). EES, ethanol extract of *S. macrocarpum*; HES, hot-water extract of *S. macrocarpum*; IL-6, interleukin-6; LPS, lipopolysaccharide; NC, negative control (without LPS stimulation); NO, nitric oxide; PC, positive control (dexamethasone 50 μg/mL and LPS stimulation); T + I, tumor necrosis factor alpha + interferon-gamma; TNF-α, tumor necrosis factor-alpha.

**Figure 4 antioxidants-11-02483-f004:**
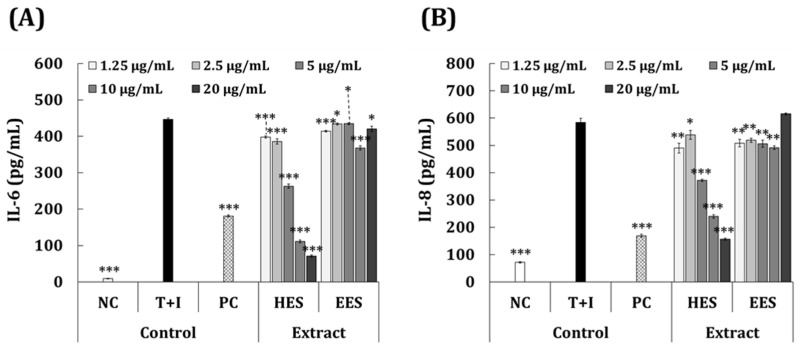
Effects of HES and EES on inhibition of inflammatory cytokines in HaCaT cells induced with T + I. HaCaT cells were incubated with 2.5–10 μg/mL of EES or HES for 24 h, then levels of (**A**) IL-6 and (**B**) IL-8 were determined using ELISAs. * *p* < 0.05; ** *p* < 0.01; *** *p* < 0.001 vs. LPS control (inflammation-induced (Student’s *t*-test). EES, ethanol extract of *S. macrocarpum*; HES, hot-water extract of *S. macrocarpum*; IL-6, interleukin-6; IL-8, interleukin-8; NC, negative control (without T + I stimulation); PC, positive control (dexamethasone 20 μg/mL and T + I stimulation).

**Figure 5 antioxidants-11-02483-f005:**
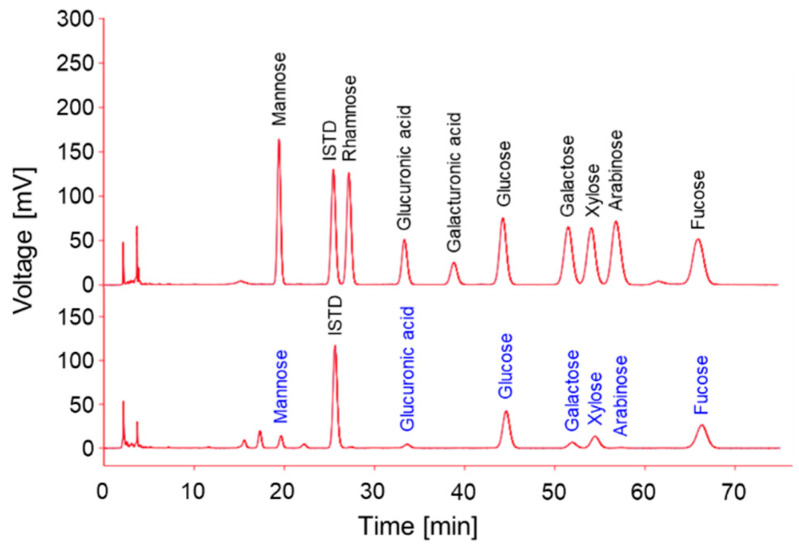
HPLC-UVD chromatogram of monosaccharide composition analysis of HES.

**Table 1 antioxidants-11-02483-t001:** Antioxidant activities of HES and EES.

Sample	ABTS	DPPH	FRAP
	(IC_50_, mg/mL)	(IC_50_, mg/mL)	(mg AA/g)
HES	1.0 ± 0.0	6.50 ± 0.3	18.8 ± 0.4
EES	16.9 ± 0.7	35.3 ± 3.1	n.d.

AA, l-ascorbic acid equivalent; ABTS, 2,2′-azino-bis (3-ethylbenzothiazoline-6-sulfonic acid); EES, ethanol extract of *S. macrocarpum*; DPPH, 2,2-diphenyl-1-picrylhydrazyl; FRAP, ferric reducing anti-oxidant power; HES, hot-water extract of *S. macrocarpum*; IC_50_, half-maximal inhibitory concentration; n.d., not determined.

**Table 2 antioxidants-11-02483-t002:** General components of HES and EES.

Sample	Total Sugar	Uronic Acid	Protein	Total Polyphenol
	(mg Gal/g)	(mg GalA/g)	(mg BSA/g)	(mg GA/g)
HES	127.6 ± 6.5	29.5 ± 0.3	41.8 ± 2.0	115.9 ± 15.3
EES	6.5 ± 3.1	2.5 ± 0.1	10.3 ± 0.4	3.9 ± 0.5

BSA, bovine serum albumin equivalent; EES, ethanol extract of *S. macrocarpum*; GA, gallic acid equivalent; Gal, galactose equivalent; GalA, galacturonic acid equivalent; HES, hot-water extract of *S. macrocarpum*.

**Table 3 antioxidants-11-02483-t003:** Monosaccharide composition of HES.

Sugar Unit	HES (mole%)
Mannose	5.0 ± 0.2
Rhamnose	–
Glucuronic acid	4.9 ± 0.3
Galacturonic acid	–
Glucose	35.7 ± 0.3
Galactose	5.7 ± 0.1
Xylose	13.7 ± 0.1
Arabinose	0.8 ± 0.0
Fucose	34.3 ± 0.2
Sum	100

## Data Availability

Not applicable.

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
