# Peer review of "Antioxidant and Anti-Inflammatory Activities of Sargassum macrocarpum Extracts"

_antioxidants, 2022, doi:10.3390/antiox11122483_

Round 1

Reviewer 1 Report

Dear authors,

After the review process, I have several comments: in the abstract, you should add numerical data and a clear aim of the paper, not only literature data; the discussion should be expanded with comments related to the bioavailability of active compounds; in this case, is relevant the relation between the bioactive potential of functional products (extracts) and bioavailability of phenolic compounds; the future applications of the results should be connected to the valorization of bioactive compounds from natural products and you should add new findings based started from separation, characterization and applications.

Best regards!

Author Response

Check the attached file

Reviewer 2 Report

Dear Authors

The manuscript is interesting and deserves consideration in Antioxidants due to the coherence of the topic.

However i have some major concerns about the following points:

1. Extraction yields: please add

2. Authors quantified mascromolecules asmajor classes of compounds, e.g. proteins, sugar, ecc..but are you about to deeply characterize them from a chromatographic-ms point of view?.

3. Statystical correlation should be reported between the antioxidant activity and the bioactive compounds found.

4. English should be revised fr typos ad grammar.

5. Introduction needs more literature background, I suggest to consider the following based on the study performed: "A Comparative Study on Phytochemical Fingerprint of Two Diverse Phaseolus vulgaris var. Tondino del Tavo and Cannellino Bio Extracts", "In vitro and in silico perspectives on biological and phytochemical profile of three halophyte species-A source of innovative phytopharmaceuticals from nature", "Calceolarioside A, a Phenylpropanoid Glycoside from Calceolaria spp., Displays Antinociceptive and Anti-Inflammatory Properties".

Author Response

Check the attached file

Round 2

Reviewer 1 Report

no other comments

Reviewer 2 Report

Dear Authors

The manuscript has been revised according to referees suggestions, now it is suitable for publication.

Best Regards